# Effects of Intraoperative Nerve Monitoring Techniques on Voice and Swallowing Disorders after Uncomplicated Thyroidectomy: Preliminary Report of a Bi-Institutional Prospective Study

**DOI:** 10.3390/jcm12010305

**Published:** 2022-12-30

**Authors:** Giuseppina Melfa, Giuseppe Siragusa, Gianfranco Cocorullo, Marianna Guzzino, Cristina Raspanti, Leone Albanese, Sergio Mazzola, Pierina Richiusa, Giuseppina Orlando, Gregorio Scerrino

**Affiliations:** 1Unit of General and Emergency Surgery, Department of Surgical Oncological and Oral Sciences, Policlinico “P. Giaccone”—University of Palermo, 90127 Palermo, Italy; 2Villa Serena Clinic—Unit of General Surgery, 90100 Palermo, Italy; 3Unit of Clinical Epidemiology and Tumor Registry, Department of Laboratory Diagnostics, Policlinico “P. Giaccone”—University of Palermo, 90127 Palermo, Italy; 4Section of Endocrinology—Department of Health Promotion Sciences Maternal and Infantile Care, Internal Medicine and Medical Specialties (PROMISE), University of Palermo, Via del Vespro 129, 90127 Palermo, Italy; 5Unit of Endocrine Surgery, Department of Surgical Oncological and Oral Sciences, Policlinico “P. Giaccone”—University of Palermo, 90127 Palermo, Italy

**Keywords:** thyroidectomy, nerve monitoring, voice disorder, aerodigestive disorder

## Abstract

Background: Voice and swallowing problems are frequently associated with thyroidectomy. Intermittent nerve monitoring (i-IONM) seems to provide a positive effect in reducing its prevalence. The aim of this study was to test the hypothesis that continuous intraoperative nerve monitoring (c-IONM) may reduce the prevalence of these disorders even further than i-IONM. Methods: This 3-arm prospective bi-institutional study compared 179 consecutive patients that underwent thyroidectomy: 56 without IONM, 55 with i-IONM and 67 with c-IONM. Neck dissections and laryngeal nerve palsies were excluded. Two questionnaires (VHI-10 for voice disorders and EAT-10 for swallowing disorders; both validated for Italian language use) were administered before and 1 month after surgery. Statistical significance was analyzed by the chi-squared test. Results: After thyroidectomy, no statistically significant differences were found in the three groups concerning EAT-10. although these symptoms seemed to be influenced by gastro-esophageal reflux. VHI-10 worsened in the “no-IONM” group compared with both i-IONM (*p* < 0.09, not quite statistically significant) and c-IONM (*p* < 0.04). Conclusion: Both i- and c-IONM improve voice quality independently of laryngeal nerve integrity. Reduced dissection and particularly restrained manipulation could explain these results, being particularly favorable for c-IONM.

## 1. Introduction

Patients undergoing thyroidectomy display voice and swallowing disorders in a limited number of cases. Usually, these complaints are related to lesions of the inferior laryngeal nerve (ILN) or the external branch of the superior laryngeal nerve (EBSLN). However, a large number of patients report these symptoms even in the absence of proven laryngeal nerve lesions [1,2,3,4,5,6]. The origin of these symptoms in formally uncomplicated thyroidectomy is complex, probably multifactorial and still unclear [7]. Mechanisms leading to laryngeal nerve injuries are well known and include stretching, ligation entrapment, thermal or electric damage, ischemia and, more rarely, clamping and transection [8]. Its incidence average, concerning ILN, was established in 2.3% of procedures for definitive and in 9.8% for transient palsies [9]. To date, EBSLN has attracted less attention. Although its lesion causes paralysis of the cricothyroid muscle, related laryngoscopic signs are mild and this complication, which ranges from 0% and 58%, remains often undiagnosed [8,10]. Laryngeal nerve palsy can lead to important speech, swallowing and/or breathing impairments and involve medicolegal claims [11].

During the last few years, the aim of reducing as much as possible the incidence of laryngeal nerve injuries has led to the exploration of tools that could improve the identification and preservation of these nerves. Intraoperative nerve monitoring (IONM) gives some advantages in identifying laryngeal nerves and preventing bilateral palsies. To date, although the use of IONM has widely increased, there is no evidence of its ability in reducing the incidence of laryngeal nerve palsy in studies that concern almost exclusively intermittent-IONM (i-IONM) [8,12,13,14,15]. A study from Silva and colleagues [16] found a positive effect of IONM in reducing post-thyroidectomy voice and swallowing symptoms, probably due to reduced manipulation and denervation following an easier ILN search. More recently, continuous IONM (c-IONM) was used in several institutions. This technique of nerve monitoring detects two main adverse events: a decrease in amplitude, because of a reduced number of neural fibers that are subjected to a stimulus, and an increase in latency, which means a reduced speed of conduction in fibers enrolled during this stimulus. Durable combined events (CE), rather than isolated amplitude or latency changes, develop as a loss of signal (LOS) in a large number of cases. Continuous IONM might determine an improvement in the prevention of ILN palsy, by enabling the surgical team to react to specific risk signals [15,17,18,19,20]. There is a lack of data concerning the favorable effects of c-IONM in reducing the adverse neural effects of thyroidectomy other than laryngeal nerve damage. In previous studies carried out at our institution, we found a significant incidence of voice and swallowing disorders associated with thyroidectomy [7,21,22]. The aim of the present study was to investigate, besides the prevention of laryngeal nerve palsy, the possible effects of IONM and particularly c-IONM in the incidence of voice and swallowing disturbances in a cohort of patients that underwent thyroidectomy.

## 2. Materials and Methods

This study was conducted on a cohort of 179 consecutive patients, recruited from two high-volume operating units for thyroid surgery, a private hospital accredited by the National Health Service and a tertiary center of the University of Palermo. The Italian Unitary Society of Endocrine Surgery (SIUEC), Association of Medical Endocrinologists (AME) and Italian Society of Endocrinology (SIE) were adopted to determine the indication for total thyroidectomy (TT) or hemithyroidectomy (HT). More specifically, conservative surgery was preferred in the case of a single or predominant lesion with a contralateral lobe free of lesions or, based on patient consent, with contralateral injuries judged clinically insignificant. Cytological patterns referred for conservative surgery belonged to Bethesda categories 2, 3 and 4, or 5 if less than 4 cm in size, in the absence of clear signs of Hashimoto’s thyroiditis or enlarged and suspected central and/or lateral cervical lymph nodes.

The following inclusion criteria were considered:–patients undergoing uncomplicated thyroidectomy (absence of transient or definitive laryngeal nerve paralysis; absence of postsurgical hypoparathyroidism, defined as a condition of hypocalcemia (Ca++ < 8 mg %) associated with low parathormone levels (PTH < 10 pg/mL) protracted beyond 6 months after surgery [23]);–complete medical records from prehospitalization to follow-up;–explicit informed verbal consent of patient for inclusion in the study.

The following exclusion criteria were considered:–linfoadenectomy of the central and/or lateral compartment;–substernal goiter;–previous neck surgery and/or irradiation.

Patients recruited for this study were allocated to one of three groups (c-IONM, i-IONM, intervention without the aid of NIM, identified as visual monitoring). In both units, it was preferred to use c-IONM, and in any case laryngeal nerve monitoring. However, in the absence of imperative indications in favor of the use of IONM [24], when the APS electrode was not available, the i-IONM technique was used; when not even the devices for this procedure were available, thyroidectomy was still performed, according to the conventionally accepted technique, with identification and sparing of the laryngeal nerves (visual monitoring).

The study was carried out on the thyroidectomies performed at the two surgical units involved in 2018. All patients were operated on by the heads of the two respective teams (G.S.^2^ and G.S.^5^). In the first 4 months of 2019, early postoperative outcomes were evaluated; the study was then discontinued because an internal audit at the university unit excluded thyroidectomy without the aid of nerve monitoring, so the “A” arm (visual monitoring) could no longer be fed further. The onset of the COVID-19 pandemic resulted in strong limitations in access to outpatient clinics for direct contact and the completion of questionnaires related to 1–2-year follow-up, as originally planned in the study design.

Thus, the characteristics of the study did not involve randomization, but it was assured by the method of recruitment, since the choice of one of the techniques of laryngeal nerve identification and monitoring was in no way dependent on an operator’s choice based on any prediction regarding specific patient characteristics.

Among the 179 patients recruited for the study, 40 underwent thyroid lobectomy (Intervention 1) and 139 underwent total thyroidectomy (Intervention 2).

In first analysis, it was evaluated whether the two types of intervention were balanced in terms of gender, age and thyroid weight. Then, the patients were divided according to the monitoring technique that they underwent A—visual monitoring, B—intermittent IONM (i-IONM) and C—continuous IONM (c-IONM).

Patients were then evaluated for voice and swallowing disorders that may have occurred after surgery. Two questionnaires were used for this purpose. They are also validated for use in the Italian language and are available online free of charge. Their purpose is to investigate the experiences of patients, before and after surgery, through the evaluation and the extent of specific disorders closely related to the voice and swallowing. For the first group of disorders, we used the Voice Handicap Index (VHI-10; Figure 1), and for the second one, we used the Eating Assessment Tools (EAT-10; Figure 2).

Both questionnaires were administered, simultaneously for each patient, before surgery (during preparation) and 1–3 months after surgery. The detected scores, from 0 to 4 for each question, were related to the optimal condition (0 = no disturbance concerning the question) and to the worst condition concerning the given question (=4). If the total score detected in the postoperative evaluation was the same compared with the preoperative one, the patient was considered not to have worsened. Conversely, any increase in score for a given question was considered to reflect the worsening of their condition.

Of note, none of the patients recruited for the study were on any treatment for gastroesophageal reflux. Notably, any treatment with proton pump inhibitors or prokinetics had been discontinued, if practiced at all, at least one month before recruitment for the study and was never practiced during the observation period. No endoscopy was performed prior to surgery. No Fiberoptic Endoscopic Evaluation of Swallowing (FEES) was performed before surgery. Ultimately, the evaluation of voice and swallowing changes was performed only by means of the respective tests.

The ethical standards of the Declaration of Helsinki (1964) and its later amendments were strictly observed.

### Statistical Analysis

A chi-square test was used to compare categorical variables, and T-Student’s test for continuous variables. Friedman’s test, which allows us to compare groups of dependent subjects, was applied to compare the same patients at different times or under different conditions (i.e., pre- versus postoperative evaluation). To verify the significance of the improvement in the scores of patients related to VHI and EAT between groups, we applied the test for equality of proportions with *p*-value = 0.05. The proportions were calculated by relating, for each group, the number of patients who showed an improvement in scores between pre and post to the number of each group.

## 3. Results

The two interventions were balanced (*p*-value = 0.7588) in terms of sex. On the contrary, they were not balanced in terms of age. In fact, the average age for Intervention 1 was 49 years and that for Intervention 2 was 53 years (*p*-value = 0.036). Concerning thyroid weight, the two interventions were not balanced; in fact, the mean weight for Intervention 1 was equal to 22 and that for Intervention 2 was equal to 57, with a *p*-value > 0.0001 (Table 1).

The differences regarding age (greater in the total thyroidectomy group) can be explained by the fact that multinodular goiter tends to be more frequent in the middle-aged, while, as far as weight is concerned, it is easy to understand that a gland affected by a single nodule is smaller than in the case of a diffuse disease.

The averages of the scores of each of the ten items in the two questionnaires are reported in Table 2, and they correspond to Interventions 1 (hemithyroidectomy) and 2 (total thyroidectomy), respectively.

Since a balance between the two types of surgery on these two parameters was not verified, we have, however, taken into account, from a statistical point of view, these differences, preferring to consider our whole sample—not distinguishing it according to the surgery performed, but evaluating the patients according to the techniques used during the surgery. Concerning the results, comparing preoperative (Figure 3a) and postoperative results (Figure 3b), the VHI score worsened in the visual monitoring group (A) (*p*-value = 0.003083). On the contrary, in the i-IONM (B) and c-IONM (C) groups, the VHI score did not worsen significantly.

We represented the results as a box plot of the differences in the total scores, for a single monitoring technique, of preoperative VHI and postoperative VHI (Figure 4).

The dashed lines in red represent the mean of these differences, and the lines in bold represent both the median value and the modal value. The position of these three values indicates the concept of symmetry/asymmetry:–for the visual monitoring technique (Group A), the difference in scores shows positive asymmetry, which means that the positive scores outweigh the negative, and this indicates a worsening of the VHI and thus a subjective and/or objective worsening of voice quality;–for the intermittent technique (Group B), the difference in scores show a slight positive asymmetry but this is not statistically significant;–for the continuous technique (Group C), the distribution of the differences between the scores is perfectly symmetrical, and the postoperative scores are identical to the preoperative ones; it shows that no worsening has occurred.

Applying the same statistical methodology (non-parametric Friedman test), we analyzed the variation in scores related to the patient’s perception of the efficiency of swallowing mechanisms, detected with the EAT questionnaire. In addition, in this case, a comparison was made between the preoperative (Figure 5a) and postoperative data (Figure 5b) based on the techniques that were used during surgery:–For i-IONM, the EAT score improved, with a *p*-value = 0.0006437.

For both i-IONM and c-IONM, the EAT score did not worsen significantly.

To explain these results, we represented, in a box plot, the differences in the total scores, by a single monitoring technique, of the EAT pre- and post-intervention (Figure 6).

The dashed lines in red represent the mean of these differences, and the lines in bold represent both the median value and the modal value. The position of these three values indicates the concept of symmetry/asymmetry:–For the intermittent technique, the difference in scores has negative asymmetry, i.e., negative scores outweigh positive scores, thus showing a statistically significant improvement;–For visual monitoring, the difference in scores is slightly negatively skewed but this is not statistically significant;–For the continuous technique, the post scores are identical to the pre scores; the distribution of the differences between the scores is therefore perfectly symmetrical, confirming that no worsening has occurred.

## 4. Discussion

Paralysis of the laryngeal nerves, particularly the inferior laryngeal nerve, is one of the most well-known and feared complications in thyroid surgery. It causes impaired phonation and swallowing and can be caused by excessive traction, resection, accidental ligation, ischemia, thermal injury caused by coagulation for hemostatic purposes, hyper- or hypothermia and hematoma [6,27,28]. In cases of bilateral paralysis, there are two different clinical expressions, depending on whether the cords are paralyzed in abduction (total aphonia and risk of aspiration of the food bolus during swallowing) or in adduction (stridor and severe obstruction of breathing, most often requiring a tracheostomy to overcome the obstacle to breathing) [1,29].

The upper aerodigestive symptoms (UADS), after thyroidectomy, could also be caused by an alteration of an anastomotic branch that connects the lower and upper laryngeal nerve, or that connects them to the cervical sympathetic chain [30].

As stated above, and as widely documented in the literature, voice and swallowing disorders are reported by many patients who, after thyroidectomy in which no complications were documented, present symptoms affecting the airway. These disorders can also substantially affect the patient’s quality of life. These symptoms, in fact, may last for a short period but may also persist or worsen over time [31,32,33,34].

In a study conducted at our institution, it was found that after total thyroidectomy, there is a constant reduction in pressure at the level of the UES. A close connection was found between incoordination of the UES, acid reflux of the proximal esophagus and phonation disorders [21].

The severity of nerve complications affecting the laryngeal nerves, especially the inferior laryngeal nerve, which can be dramatic if the lesion is bilateral, has led to the exploration of methods aimed at ensuring safety as much as possible. Recently, techniques for the instrumental detection of laryngeal nerves have been introduced. These techniques have arisen from the experience gained in the second half of the twentieth century, at the end of the century and at the beginning of the current one, and they have been perfected until the development and standardization of intraoperative monitoring of laryngeal nerves (IONM) [35,36,37,38,39].

It has been demonstrated that the technique of IONM is effective in the early detection of laryngeal nerve injury, and consequently has led to a significant reduction in the most serious complication of thyroidectomy (bilateral paralysis of the inferior laryngeal nerve), but its effectiveness in preventing even monolateral lesions of the nerve remains controversial [13,40].

The evolution of IONM is the technique of continuous neuromonitoring (c-IONM), which has allowed an even further increase in safety in the execution of this intervention by increasing the level of prevention [41,42,43,44]. In fact, the continuous stimulation of the nerve, during the execution of the intervention, allows the surgeon to obtain constant feedback as an indication of the suffering of the nerve or its anastomoses that leads to the suspension of the potentially damaging dissection or maneuver, thus avoiding irreversible nerve alterations [45,46].

This study confirms the importance of nerve monitoring in increasing the safety of the procedure and demonstrates, in particular, that the strict respect of the laryngeal nerves during thyroidectomy has additional advantages that result, beyond the nerve complications, in a reduction in aerodigestive symptoms related to thyroidectomy even in the absence of complications. These results are not completely explainable in light of current findings; however, some hypotheses can be made. The optimal preservation of the laryngeal innervation as a whole, which is in turn linked to the strict control of the manipulations and tractions that the technique undoubtedly causes, could be an acceptable explanation. In our study, these good results are clearer in the field of vocal complaints. Swallowing disorders are also favorably influenced by the monitoring techniques, although, in this sense, the difference between i-IONM and c-IONM was less clear and not significant.

It should be remembered, as demonstrated by previous studies carried out by our team as well as other groups [7,21,22,33], that swallowing disorders are strongly influenced by the function of the upper esophageal sphincter, whose regulatory mechanisms could be more complex and not exclusively limited to nervous control.

The results of this study are the first to demonstrate a relationship between the use of IONM and the prevention of aerodigestive disorders independently of laryngeal nerve impairment. However, it is burdened by several limitations, such as the absence of randomization in the strict sense and the strong imbalance of the three groups in terms of patients enrolled, so that patients undergoing thyroid lobectomy, on the one hand, and those who were monitored with c-IONM, on the other, were scarce. It would have been appropriate to conduct a study with randomization of the indication of each type of treatment. Moreover, in patients who are candidates for surgical treatment, it is important to take into account the diversity of results depending on inter-operator variability. However, we would like to specify that both surgeons, who operated on the whole sample (G.S2. and G.S5.), are considered “high-volume surgeons” in accordance with the literature data [47,48]. This may constitute a partial reduction in the variability of inter-operator outcomes, as the risk of surgery failing to satisfy quality standards can be considered low. A risk of selection bias, although not burdened by intentionality, may also be highlighted. On the other hand, statistical significance may not have been reached for some parameters evaluated, precisely because of the limited sample, followed for a short postoperative period. It would be advisable to carry out a study with randomization and a larger sample.

Moreover, no examinations (endoscopy) to evaluate the glottis or FEES before surgery were performed. These can also be considered limitations of the study.

Finally, it should be noted that the results were evaluated only in the short term, as the onset of the pandemic prevented remote follow-ups, causing a number of patients to be lost to follow-up.

We therefore believe that this line of studies deserves further verification and investigation through an expansion of the sample to be examined and, possibly, with more advanced instruments for the measurement of alterations in laryngeal and esophageal motility as a whole.

The specific advantages highlighted in this study in favor of IONM techniques add to what has already been widely demonstrated in the literature about this technique, which actually reduces bilateral vocal fold paralysis to near-zero probability and, in the specific case of c-IONM, seems capable of further containing the overall risk of paralysis [49,50]. By minimizing critical maneuvers, c-IONM could become a new gold standard in most thyroid surgery scenarios [51].

## 5. Conclusions

In conclusion, the c-IONM technique appeared, in our experience, to be simple, safe and reproducible. An immediately obvious advantage is the early identification of possible inferior laryngeal nerve stress, manifested by electromyographic curve changes in terms of latency and/or impedance, identified at the very moment that it occurs. Based on the results obtained in this study, we believe that continuous monitoring should be performed in all cases of thyroid surgery.

This study did not include a cost analysis; however, continuous monitoring is the only tool currently available that can improve the rate of recurrent paralysis, compared with the low cost of APS and the few minutes required for its installation. These results produce undoubted changes in daily clinical practice.

The results of our study allow us to state that a delicate and meticulous surgical technique, a systematic visualization of the laryngeal nerve along the entire cervical course, up to its entry into the larynx, and, finally, the aid of c-IONM can not only reduce the incidence of laryngeal nerve injury but also determine advantages concerning aerodigestive disorders. The data acquired with our study, to be considered initial, encourage further verification through studies with larger population samples to be evaluated over longer periods of time.

## Figures and Tables

**Figure 1 jcm-12-00305-f001:**
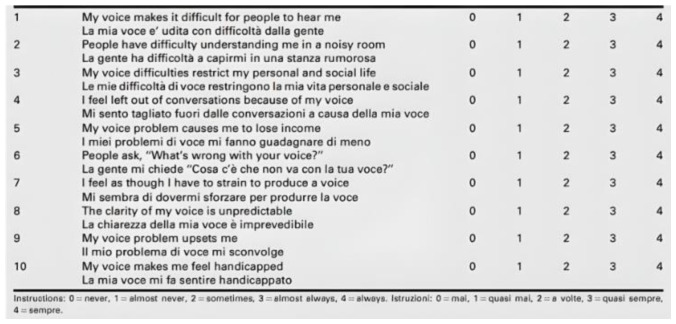
Italian VHI-10 Questionnaire [25].

**Figure 2 jcm-12-00305-f002:**
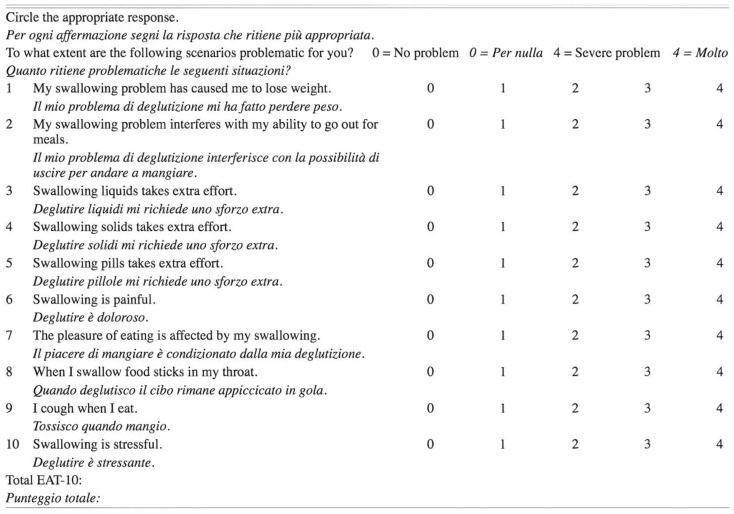
Italian EAT-10 Questionnaire [26].

**Figure 3 jcm-12-00305-f003:**
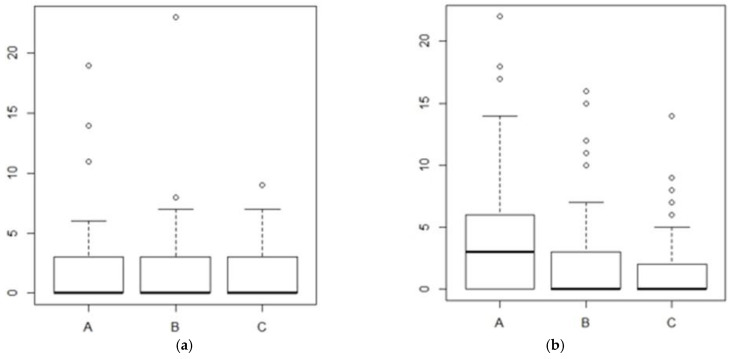
Comparison between preoperative (**a**) and postoperative VHI data (**b**) based on monitoring techniques. A = visual technique; B = intermittent nerve monitoring; C = continuous nerve monitoring.

**Figure 4 jcm-12-00305-f004:**
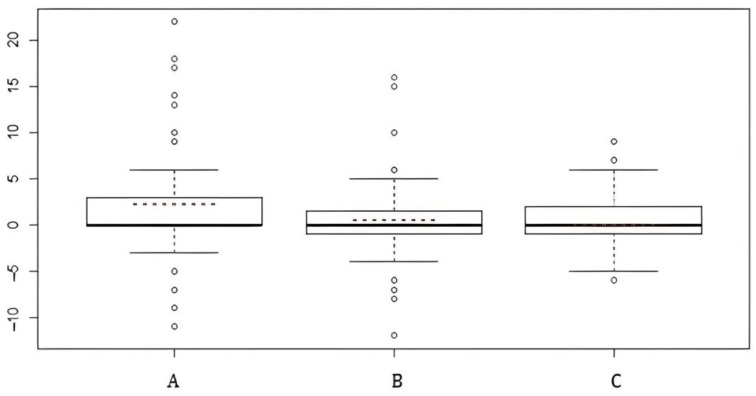
Differences in the total scores, for single monitoring technique, of preoperative VHI and postoperative VHI. A = visual technique; B = intermittent nerve monitoring; C = continuous nerve monitoring.

**Figure 5 jcm-12-00305-f005:**
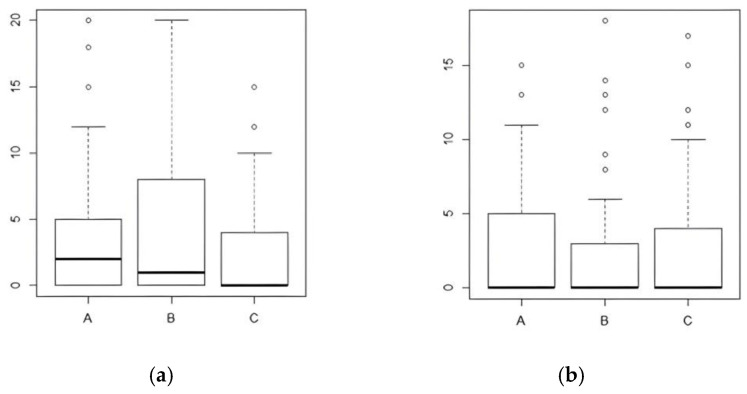
Comparison between preoperative (**a**) and postoperative EAT data (**b**) based on monitoring techniques. A = visual technique; B = intermittent nerve monitoring; C = continuous nerve monitoring.

**Figure 6 jcm-12-00305-f006:**
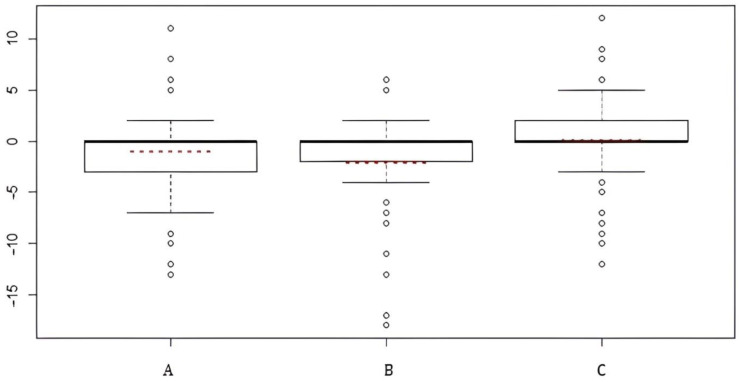
Differences in the total scores, by single monitoring technique, of EAT pre- and post-intervention. A = visual technique; B = intermittent nerve monitoring; C = continuous nerve monitoring.

**Table 1 jcm-12-00305-t001:** Results.

		NIM			
**VARIABLES**	**INTERVENTION 1**		**INTERVENTION 2**	**Total**	*p*-value
Male	9		26	35	
Female	31		113	144	0.7588
**Total**	40		139	179	
**VARIABLES**	**INTERVENTION 1**		**INTERVENTION 2**		*p*-value
**Age**	49		53		0.036
**VARIABLES**	**INTERVENTION 1**		**INTERVENTION 2**		*p*-value
**Thyroid weight**	22		57		>0.0001
**VARIABLES**	**Visual**	**Intermittent**	**Continuous**	**Total**	*p*-value
Male	10	12	13	35	
Female	45	44	55	144	0.9052
**Total**	55	56	68	179	
**VARIABLES**	**Visual**	**Intermittent**	**Continuous**	*p*-value	
**Age**	55	53	53	0.7553	
**VARIABLES**	**Visual**	**Intermittent**	**Continuous**	*p*-value	
**Thyroid weight**	35	40	43	0.5736	

**Table 2 jcm-12-00305-t002:** Averages of the scores of the ten items in patients undergoing hemithyroidectomy and total thyroidectomy.

Hemithyroidectomy (Intervention 1)		D1	D2	D3	D4	D5	D6	D7	D8	D9	D10
Visual	**VHI PRE**	0.44	0.37	0.19	0	0.12	0.25	0.37	0.31	0	0
Visual	**VHI POST**	0.75	0.81	0.44	0.19	0.12	0.44	1.19	1.19	0.31	012
Visual	**EAT PRE**	0	0	0.57	0.69	0.56	0.31	0.12	0.44	0.06	0.06
Visual	**EAT POST**	0	0.06	0.43	0.43	0.43	0.12	0	0.27	0	0.06
Intermittent	**VHI PRE**	0.14	0.21	0.07	0.07	0	0.28	0.57	0.64	0.07	0.07
Intermittent	**VHI POST**	0.21	0.64	0.21	0.21	0.07	0.64	1.29	1.07	0.5	0.07
Intermittent	**EAT PRE**	0.07	0.21	0.35	1,07	0.79	0.21	0.43	0.07	0.43	0.43
Intermittent	**EAT POST**	0	0.07	0.43	0.64	0.57	0.14	0.14	0.07	0.43	0.5
Continuous	**VHI PRE**	0	0.3	0	0	0	0.3	0	0.3	0	0
Continuous	**VHI POST**	0	0.5	0.1	0.2	0.2	0.5	0.2	0.4	0.1	0
Continuous	**EAT PRE**	0	0.4	0.7	1.44	0.9	0.1	0.3	1	0.3	0.4
Continuous	**EAT POST**	0	0.2	0.5	0.3	0.3	0	0.1	0.3	0.2	0.2
**total thyroidectomy (intervention 2)**											
Visual	**VHI PRE**	0.15	0.20	0.31	0.33	0.31	0.33	0.36	0.44	0.23	0.15
Visual	**VHI POST**	0.13	460.	0.18	0.36	0.28	0.49	0.36	0.51	0.25	0.18
Visual	**EAT PRE**	0.02	0.02	0.74	0.92	0.79	0.44	0.41	0.59	0.44	0.28
Visual	**EAT POST**	0	0.05	0.56	0.7	0.71	0.43	0.23	0.64	0.33	0.26
Intermittent	**VHI PRE**	0.21	0.17	0.07	0.07	0.04	0.02	0.31	0.29	0.07	0.02
Intermittent	**VHI POST**	0.17	0.26	0.21	0.12	0.02	0.07	0.48	0.38	0.07	0
Intermittent	**EAT PRE**	0.07	0.12	0.62	0.98	0.52	0.14	0.07	0.55	0	0.09
Intermittent	**EAT POST**	0.04	0.07	0.24	0.29	0.15	0	0.07	0.07	0	0.05
Continuous	**VHI PRE**	0.05	0.29	0.19	0.07	0.14	0.43	0.14	0.47	0.02	0
Continuous	**VHI POST**	0.09	0.31	0.19	0.15	0.15	0.33	0.14	0.28	0.05	0.02
Continuous	**EAT PRE**	0.12	0.15	0.38	0.46	0.46	0.22	0.15	0.33	0.21	0.14
Continuous	**EAT POST**	0.15	0.22	0.4	0.45	0.4	0.29	0.19	0.38	0.33	0.03

Patients undergoing hemithyroidectomy (intervention 1) and total thyroidectomy (intervention 2). The mean scores for each individual question, for each questionnaire (VHI and EAT) were calculated in the three groups: visual, intermittent and continuous.

## Data Availability

The datasets generated during and/or analyzed during the current study are available from the corresponding author on reasonable request.

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
