# Peer review of "Effects of Intraoperative Nerve Monitoring Techniques on Voice and Swallowing Disorders after Uncomplicated Thyroidectomy: Preliminary Report of a Bi-Institutional Prospective Study"

_jcm, 2022, doi:10.3390/jcm12010305_

Round 1

Reviewer 1 Report

The method of allocation of patients in each study arm is not clearly assigned. It would be advisable to have randomized the indication of each type of treatment.

In patients who are candidates for surgical treatment, it is important to take into account the diversity of results depending on interoperator variability, to avoid bias. The method of allocation of patients in each study arm is not clearly assigned. It would be advisable to have randomized the indication of each type of treatment.

In patients who are candidates for surgical treatment, it is important to take into account the diversity of results depending on interoperator variability, to avoid bias.The method of allocation of patients in each study arm is not clearly assigned. It would be advisable to have randomized the indication of each type of treatment.

In patients who are candidates for surgical treatment, it is important to take into account the diversity of results depending on interoperator variability, to avoid bias. Have you taken this variability into account? How?

It would be advisable to carry out a study with randomization and a larger sample.

Based on the results obtained in your study, do you think continuous monitoring should be performed in all cases of thyroid surgery?
Justify your answer in clinical terms and in terms of efficiency (cost analysis).
What change in daily clinical practice have generated the findings of your study?
How do you think you can improve the consistency of the results in relation to interoperator variability?
In which cases do you not believe the use of continuous monitoring versus intermittent monitoring is justified?
In which cases do you think it is justified not to carry out monitoring?

Author Response

We thank the reviewers for their hard work, which enabled us to greatly improve the quality of the manuscript. Below, we list the changes we felt we should make based on their suggestions

1) Title change: Although not specifically suggested by the reviewers, we felt we should modify the title by emphasizing that this is an observation-only study, in which randomization is lacking in both patient and surgeon inclusion.  Both surgeons should still be considered "high volume" according to the standards of the literature.

Reviewer 1

2) Questions 1-3: We have extensively edited the manuscript in Discussion, lines 303-307 (of the new manuscript), further specifying that the patients enrolled in the study were not randomized and neither were the surgeons: this is its limitation. Regarding the surgeons, however, we clarified that all patients were operated on by the two heads of the teams, both of which are to be considered high-volume according to widely shared literature (References 46, 47). In lines 106-107 (Materials and Methods) were also indicated the two surgeons.

3) Question 4: We added this sentence in lines 314-315.

4) Question 5-6-7: We added in Conclusions (lines 336-341): Based on the results obtained in this study, we think continuous monitoring should be performed in all cases of thyroid surgery. We added a sentence concerning costs and advantages of c-IONM The sentence inserted clarified the role of APS in changes of daily clinical practice.

7) Question 8: “How do you think you can improve the consistency of the results in relation to interoperator variability”? We think the answer has already been given with the sentence inserted in lines 304-310 of the discussion

8) Questions 9-10: “In which cases do you not believe the use of continuous monitoring versus intermittent monitoring is justified”? and  “In which cases do you think it is justified not to carry out monitoring”? The questions posed by the reviewer were not directly addressed as primary endpoints in our study, nor does the literature provide firm data in this regard. What we state in sentence line 329-330, supported by ref. no. 50, allows us to argue that, ultimately, there are no real limitations to the use of c-IONM.

Reviewer 2 Report

How was gastro-esophageal reflux evaluated? Was any treatment given before or after surgery?

Was endoscopy performed on all patients before the surgery to exclude any glottic pathology before the surgery?

"Patients were then evaluated on voice and swallowing disorders that may have occurred after surgery 121" Were these functions evaluated also before surgery?

Was the FEES done to the patients that had swallowing disorders post-op in order to determine the cause of dysphagia?

The scores of the two questionnaires before and after the surgery should be demonstrated in a table. 

Author Response

We thank the reviewers for their hard work, which enabled us to greatly improve the quality of the manuscript. Below, we list the changes we felt we should make based on their suggestions

1) Title change: Although not specifically suggested by the reviewers, we felt we should modify the title by emphasizing that this is an observation-only study, in which randomization is lacking in both patient and surgeon inclusion.  Both surgeons should still be considered "high volume" according to the standards of the literature.

Reviewer 2

1) In lines 145-150 we answered the first four questions: “…none of the patients recruited for the study were on any treatment for gastro-oesophageal reflux. Notably, any treatment with proton pump inhibitors or prokinetics had been discontinued, if practiced at all, at least one month before recruitment for the study and was never practiced during the observation period. No endoscopy was performed prior to surgery. No Fiberoptic Endoscopic Evaluation of Swallowing (FEES) was performed before surgery. Ultimately, the evaluation of voice and swallowing changes was done only by means of the respective tests”.

2)In lines 176-178 we responded to the suggestion to include a table summarizing the scores of the two questionnaires (which for greater readability we have doubled according to the intervention made), introducing them with an additional sentence.